# Deep Symbolic Optimization for Electric Component Sizing in Fixed Topology Power Converters

**Ruben Glatt**[1*], **Felipe Leno da Silva**[1*], **Van-Hai Bui** [2], **Can Huang** [1], **Lingxiao Xue** [3], **Mengqi Wang** [2], **Fangyuan Chang** [2], **Yi Lu Murphey** [2], **Wencong Su** [2]

[1] Lawrence Livermore National Laboratory
[2] University of Michigan, Dearborn
[3] Oak Ridge National Laboratory
glatt1,lenodasilva1@llnl.gov

## Abstract

Power converters (PC) are a major component in any current electronic hardware device. The development and design is usually guided by expert knowledge and heavily relies on human intuition and experience. The process is a very time consuming and costly activity and it is generally hard to improve upon current designs. As a first step towards autonomous PC design, we are here proposing a new framework for the sizing of components for fixed topology PCs based on given design requirements. To this end, we developed surrogate models for rapid evaluation of new topologies and adapt the deep symbolic optimization (DSO) framework to generate new topologies guided by a reinforcement learning training signal. In an empirical evaluation, we show that our DSO based approach is able to find the optimal configuration for all investigated topologies, while reducing the learning time by at least a factor of 100 compared to popular RL algorithms.

## Introduction and Background

Power converters (PC) are an integral component in today's electronic devices and play a major role in technological development. With the increasing rate of electrification and digitalization, the PC market is projected to grow continuously over the next years. This will mainly be driven by developments in the energy and power sector, as well as by massive growth of the aerospace and robotics industries, having implications on many aspects of our daily lives (Vertical 2021).

Designing efficient PCs is an expensive task, requiring human experience and costly testing and simulation. The basic building blocks are electronic components such as resistors, capacitors, inductors, diodes, and switching devices. The complexity and difficulty come from the combination of these blocks in highly interconnected circuits. As small changes can lead to inefficiencies, the whole process is time-consuming, inefficient, and labor intensive. This effect is intensified by application-specific considerations, such as cost, thermal, or packaging constraints. Nevertheless, the state-of-the-art process is still heavily reliant on human experts to select the optimal topology and search for design parameters based on the expert's experience and intuitions.

On the other hand, Machine Learning (ML) as a tool to automate general processes has started to make an increasing impact. Deep Learning (DL) (LeCun, Bengio, and Hinton 2015) approaches have led to a number of breakthroughs in computer vision (Voulodimos et al. 2018), natural language processing (Otter, Medina, and Kalita 2020), and recommendation systems (Batmaz et al. 2019). The representational generalization abilities of DL also led the way to new developments in autonomous decision making based on Deep Reinforcement Learning (DRL) (Arulkumaran et al. 2017; Sutton and Barto 2018). DRL has shown to learn super-human control in areas such as board games (Silver et al. 2016), Atari game playing (Mnih et al. 2015; Glatt et al. 2020), electric vehicle control (Pettit et al. 2019; Silva et al. 2019), robotics (Levine et al. 2016), energy applications (Zhang, Zhang, and Qiu 2019; Liang et al. 2021), chip design (Mirhoseini et al. 2020) and many other domains.

More recently, ML has been employed into power electronics system design with the goal of speeding up the process. Dragičević, Wheeler, and Blaabjerg (2018) establish a functional relationship between design parameters and reliability metrics, and use them as the basis for optimal design in a grid-connected photo-voltaic converter case study. Wang et al. (2018) leverage reinforcement learning to automatically search circuit parameters and evaluate their methods on two different trans-impedance amplifiers circuits. Their approach is able to design circuits with better performance than random search, Bayesian Optimization and human experts. In an extension of their work, Wang et al. (2020) propose GCN-RL Circuit Designer, an RL agent based on Graph Convolutional Neural Network (GCN) architecture to transfer knowledge between different technology nodes and topologies for transistor sizing. Their experiments demonstrate that the method can achieve better results than others through knowledge transfer and enables more effective and efficient transistor sizing and design porting.

Despite the efforts in those works, autonomous development (or recommendation) of efficient PC topologies remains a challenging and unresolved research field. Given the successful recent history of ML-powered design specification, we propose a method to support the development of

---
*These authors contributed equally.

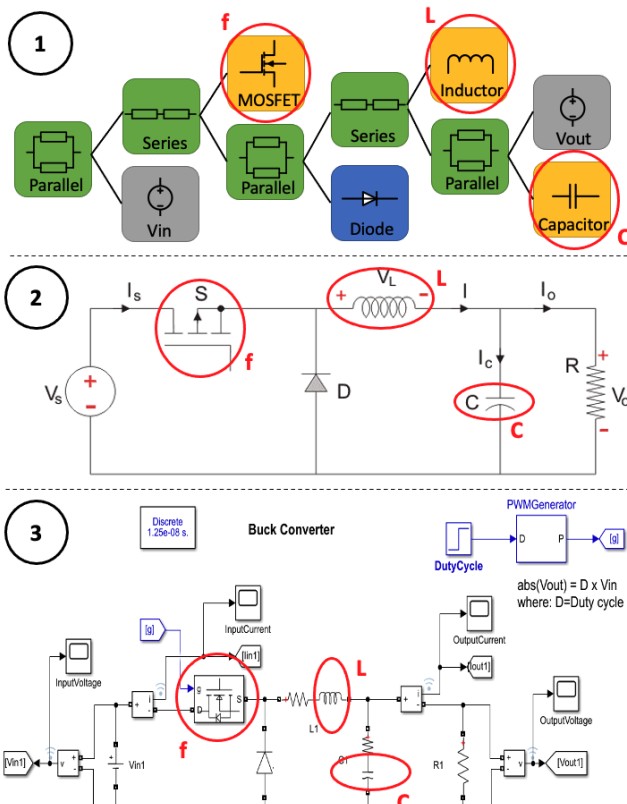

Figure 1: The figure shows different representations of the same converter topology with varying detail and structure: 1) topology description in a tree structure as generated with our framework, 2) simple circuit diagram derived from the tree, and 3) full model generated using a simulation software.

PC topologies as part of an intelligent system for automatic selection of DC-DC converters (Wang et al. 2022). The main contributions of this paper are (1) the representation of PC topologies as a tree structure, (2) an RL based framework to optimize component selection for a desired topology, and (3) an empirical evaluation showing that our method is useful for this task.

## Power Converter Representation

Figure 1 illustrates how electric circuits are interchangeably represented in different levels of abstraction by our approach. In this example, we use a step-down converter, or buck converter, to regulate a voltage level, designated as $V_{in}$, to a lower voltage of $V_{out}$. We depict the buck converter because it can can be ubiquitously found in almost all modern electronics devices and is also one of the fundamental topologies in power electronics. In Figure 1 (1), a high-level representation of the circuit as a discrete token tree is shown. Tokens can either represent how components are linked (serial or parallel connections) or actual components. This token tree representation is convenient for manipulation by our learning algorithm, described in the next

section. By traversing the tree from left to right, we can uniquely recover an electric circuit from the token tree, illustrated in Figure 1 (2). This simple circuit model can be fully implemented through physical components or by using commercial simulation software, as in Figure 1 (3).

## Deep Symbolic Optimization (DSO)

Given the circuit token tree representation discussed in the previous section, we propose to use the *Deep Symbolic Optimization* (DSO) framework to learn how to optimize circuits. DSO is a framework that allows to explore the space of possible solutions that optimize hierarchical, variable-length discrete objects under a black box performance metric (Petersen et al. 2021; Landajuela et al. 2021a,b). DSO refers to solutions as *programs* $\tau = [\tau_0, \tau_1, \ldots, \tau_{|\tau|}]$ which are generated by the sequential combination of functional *tokens* that are sampled from a token library $\mathcal{L} = \{\tau^1, \ldots, \tau^t\}$ under the consideration of prior knowledge and logical constraints. The programs form an expression tree with tokens that present internal nodes, which are operators, and terminal nodes which are constants or the input variables of the dataset. The programs are evaluated based on a reward function which indicates how good a specific program is.

DSO is composed of a sequence generating neural network, is based on a Recurrent Neural Network (RNN) architecture. The RNN provides a parameterized distribution over all tokens in the library $p(\tau|\theta)$ with parameters $\theta$. The RNN is trained using a batch of programs $\mathcal{T}$ and backpropagating the gradients of a defined loss function, naturally, intended to learn to generate programs that optimize the reward:

$$\mathcal{L}(\theta) = \frac{1}{\varepsilon|\mathcal{T}|} \sum_{\tau \in \mathcal{T}} (R(\tau) - \tilde{R}_\varepsilon) \nabla_\theta \log p(\tau|\theta) \mathbf{1}_{R(\tau) > \tilde{R}_\varepsilon} \quad (1)$$

where $\varepsilon$ determines the degree of risk-seeking and $\tilde{R}_\epsilon$ is the empirical $(1 - \varepsilon)$ reward quantile of $\mathcal{T}$. *Risk-seeking* here refers to degree in which DSO optimizes the *best-case* scenario, i.e., is concerned only in the best solutions for the problem found so far. This is unlike typical RL methods, which optimize the average performance.

DSO is especially well-suited for our problem at hand, since: (i) circuits can be represented in a "symbolic" way through tokens; and (ii) we are interested in finding the *best* circuit for a particular setting. In the next section, we present our modeling of circuit optimization as a symbolic optimization problem.

## Topology Generation framework

Our goal is the development of a ML-driven system that supports the design of new PCs, by quickly optimizing the component selection constrained to a desired circuit topology. We use DSO for the problem at hand, where a hierarchical specification (the PC) of discrete objects (the electronic components) is needed.

On a high level, the framework is composed of a topology generator and a policy evaluator. While the former generates topology parameters that seem promising, the latter evaluates the generated topologies and produces a training signal for the generator to improve topology generation over

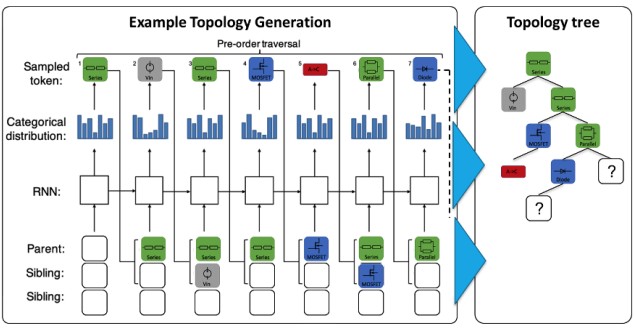

Figure 2: Topology generation process based ob the $DSO$ framework.

| | Buck.r | Buck | Boost | BuckBoost |
|---|---|---|---|---|
| $V_{in}$[V] | 48 | 48 | 5 | 20 |
| $V_{out}$[V] | 5 | 5 | 15 | 10 |
| $P_{out}$[W] | 20 | 20 | 20 | 5 |

Table 1: Fixed design requirements for the selected topologies.

time. The performance indicator for our purposes is the efficiency of the topology. The evaluation process for a single topology is usually computationally expensive in commercial simulation software. As we frequently need to evaluate new topologies, the long time taken for computing the efficiency was a significant obstacle. Therefore, we developed a surrogate model to quickly estimate the efficiency which allows us to iterate over designs quickly.

Note that DSO generates different samples from the distribution that might have different tree structures of different size; thus, the search space is inherently both hierarchical and variable in length. In order to improve search and ensure that valid circuits are generated, pre-defined constraints can be applied. For example, one can fix certain positions in the tree or impose length or architecture demands on the generated topologies to integrate human knowledge. These constraints can guide if the learning is more focused on generating completely new topologies or modifications of existing ones. Eventually, the framework produces a number of topologies that meet the user defined requirements and proposes them to the user to make a final selection based on user preference or just suggests the best performing topology. DSO produces a Hall of Fame (HoF) containing the best performing topologies which makes this process trivial.

In this work, we are only considering topologies defined by the user, which allows us to focus the learning on the sizing of the electronic components. As described earlier, we can represent topologies as a tree structure. By fixing the topology, we can define constraints on the components that the generator can place at each node in the tree. In Figure 2, we can see the process of generating a topology in more detail (shown token are randomly selected for demonstration only). In the left box, we can see the columns that represent the token selection process at each step that sequentially build the topology tree. The input to the RNN is comprised of a parent node and the sibling nodes of the token that will be selected next. Initially, no tokens have been drawn so that neither parent nor sibling nodes exist. The *categorical distribution* is a probability distribution over all tokens in the library considering any prior knowledge or constraint and will change at every step depending on the current sequence. The selected token then becomes a parent or sibling node in the next step until all branches end in terminal tokens.

## Experiments

We investigated the benefits of our proposed framework through an empirical evaluation on fixed topology component sizing. For this purpose, we focus on three PC topologies: (1) Buck, (2) Boost, and (3) BuckBoost. Those topologies have in common that they are generally built from the same type of components consisting of (1) inductances, (ii) capacitances, and (iii) pulse width modulators (PWMs); allowing us to use an identical tokenset for all topologies. Design requirements are other important parameters of each topology and remain fixed as shown in Table 1.

Generated topologies are evaluated by an estimate of their efficiency. Although the efficiency can be accurately estimated through the use of a simulation software (such as Modelica (Urquía et al. 2018)), simulation times can take up to several minutes, which makes it unfeasible to run simulations as part of a learning process. We overcome this issue by developing a surrogate model that approximates full circuit simulations with a much faster execution speed, enabling to evaluate circuits immediately. We use an individual surrogate model for each topology that has been previously trained to evaluate the performance of each of the aforementioned topologies. The surrogate models are based on a deep neural network architecture and trained on data generated by Matlab/Simulink. For every model, we generated about 30,000 input combinations of switching frequency, inductance, and capacitance and their respective power efficiency. The model architecture is a simple feed-forward neural network with 10 topology parameters as input, 512 nodes in a hidden layer, and a single output node representing the efficiency based on the input parameters.

In all experiments, we compare the DSO framework against *Bruteforce*, *Q-Learning* (Watkins and Dayan 1992), and *DDPG* (Lillicrap et al. 2015). Bruteforce searches naively evaluating all possible component combinations among a discretized set of possible components. It should always find the best result for the discrete tokenset which is the same in DSO and Q-Learning, so it serves as a upper baseline for our algorithm. DDPG, on the other hand, operates in continuous space (in a given range) and might be able to achieve a better efficiency after training than the discrete algorithms. However, notice that DDPG can possibly propose circuits composed of components that are not commercially available, due to the continuous space of possible parameter values. Q-Learning and DDPG were trained by finetuning a given initial setting of the parameters and adjusting each component to increase the efficiency over a number of steps. Differently, in the DSO framework, we are sampling each token sequentially using the RNN and additionally constrain the sample process by allowing the algorithm to only sam-

| | Buck.r | Buck | Boost | BuckBoost |
|---|---|---|---|---|
| **BruteForce** | **0.955** | 0.976 | **0.995** | **0.988** |
| **Q-Learning** | **0.955** | 0.975 | **0.995** | 0.954 |
| **DDPG** | 0.951 | **0.977** | 0.992 | 0.969 |
| **DSO** | **0.955** | 0.976 | **0.995** | **0.988** |

Table 2: Highest efficiencies and running time for each algorithm in our empirical evaluation for the three PC topologies (bold=best). DSO finds optimal settings for all topologies.

ple the right type of token for each position in the topology tree. For our experiments, this means that one token of each type, switching frequency $f_{PWM}$, inductance $I$, and capacitance $C$ will be sampled, while the rest of the tree remains constant at their respective positions.

We performed two rounds of experiments. In the first round, we only considered the Buck converter and also only used a reduced tokenset (19 tokens) in the DSO framework. The results of this experiment are denoted as *Buck.r* in tables and figures. In Table 2 we can see that the reduced tokenset in general achieves lower efficiency than the full tokenset, however, we can see that DSO manages to find the best solution as indicated by Bruteforce. Figure 3 shows the efficiency of the top 100 topologies, the *Hall of Fame (HoF)*, found by our algorithm during the training process for all experiments. The top graph shows the results for Buck.r. As an interesting side effect, the HoF graph revealed the component most responsible for improving the efficiency in this example. The observable steps in the curve have an overwhelmingly common component which is the frequency token that shows increases with rising frequency, posing an interesting conclusion that could be used for further improvements and increase in human knowledge.

The second round of experiments considered all topologies and an extended tokenset (40 tokens). In Table 2 we can see that DSO recovers the best solution for all topologies as indicated by Bruteforce. Q-Learning fails to find the best performing solution reliably and DDPG only once outperforms the Bruteforce approach but trails slightly in the other settings. As all learning approaches show good performance, only DSO consistently recovers the optimal component selection for the different PCs. Another great benefit is that while a complete training run in Q-Learning and DDPG takes well over 30 minutes for each topology to converge to good solutions, DSO reduces the training time by a factor of more than 100 to less than 20 seconds. The HoF and the best performing component selection for each topology as found by DSO are shown in Figure 3.

## Discussion & Conclusion

In this paper, we proposed a framework to support the design and development of PCs. By leveraging RL-based symbolic optimization, we represent PC topologies as sequences of discrete components and autonomously learn how to optimally size the components in a fixed topology. We also developed a simple surrogate model to quickly estimate topology quality and overcome the restrictions of long computing cycles for efficiency determination. We presented an empiri-

cal evaluation that shows that our method finds optimal solutions by using much less computation time. For future work, we plan to extend our framework to generate new converter topologies with fewer architecture restrictions by integrating more human knowledge in form of priors and constraints to guarantee validity of generated topologies.

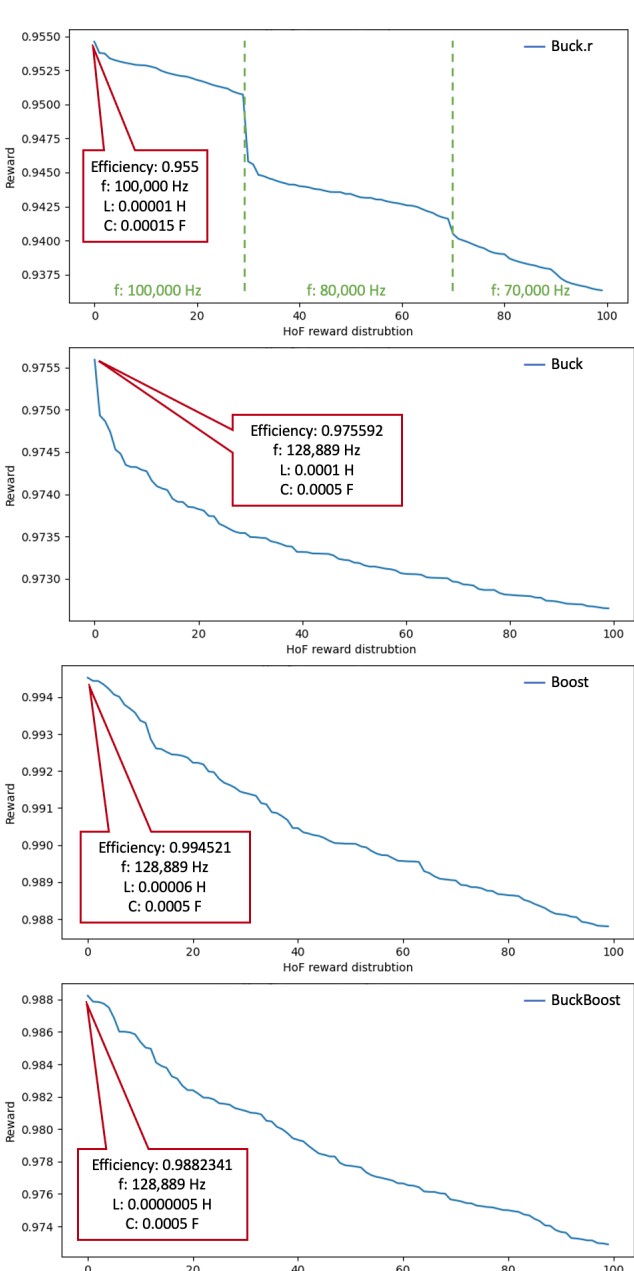

Figure 3: Overview of the top 100 generated topologies by DSO ordered by reward (efficiency) for (from top to bottom): (a) Buck power converter with reduced tokenset, (b) Buck power converter with full tokenset, (c) Boost power converter with full tokenset, and (d) Boost power converter with full tokenset.

## Acknowledgments

The information, data, or work presented herein was funded in part by the Advanced Research Projects Agency-Energy (ARPA-E), U.S. Department of Energy, under Award Number DE-AR0001219. The views and opinions of authors expressed herein do not necessarily state or reflect those of the United States Government or any agency thereof. The work of Ruben Glatt, Felipe Leno da Silva, and Can Huang was performed under the auspices of the U.S. Department of Energy by Lawrence Livermore National Laboratory under contract DE-AC52-07NA27344. Lawrence Livermore National Security, LLC. LLNL-CONF-828961.

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
