# OpenReview forum: "Deep Symbolic Optimization for Electric Component Sizing in Fixed Topology Power Converters"
_AAAI.org/2022/Workshop/ADAM — AAAI 2022 Workshop ADAM_

### Official Review · Reviewer_Y2bm · 2021-11-27
**Review for deep symbolic optimization for electric component**

**Rating:** 7
**Confidence:** 3

**Review:**

Summary: This paper applied deep symbolic optimization method to size the electric component in fixed topology power converters. Specifically, the authors presented a new framework for the sizing of components for fixed topology PCs based on given design requirements by leveraging a RL training signal, which has shown the superiority over the traditional methods based on domain knowledge that could be time-consuming in the design. The application in this work seems to be interesting and novel in the design of PCs. The empirical results look promising for different fixed topologies. While the paper is well written and easy to follow, I think the paper can be improved by taking into account the following aspects. First, the authors should give a little more detail on the RL framework though I can understand it might be due to the limit of the space. In the current draft, it is a bit difficult to understand the DSO based on the RL training signal. Second, it would be better to see some concrete results on the comparison of learning time, though Table 2 has shown the improvement in terms of efficiency.

---

### Official Review · Reviewer_nJZc · 2021-12-02
**Deep Symbolic Optimization for PC Design**

**Rating:** 7
**Confidence:** 4

**Review:**

This paper proposes a RL-based optimization framework of the components of a power converter topology. Interestingly, they represent the PC topology as a program tree, and use a RNN to populate the tokens in the tree while maximizing a reward function. The paper is nicely written and is easy to follow.

Below I list suggestions/comments to further improve the paper:
(1) Many details, such as the RNN model, RL training protocol, number of tokens, accuracy of the surrogate model, are currently missing, which can be added to the Appendix.